# The Influence of Competitive Level on Stretch-Shortening Cycle Function in Young Female Gymnasts

**DOI:** 10.3390/sports10070107

**Published:** 2022-07-06

**Authors:** Sylvia Moeskops, Jason S. Pedley, Jon L. Oliver, Rhodri S. Lloyd

**Affiliations:** 1Youth Physical Development Centre, Cardiff School of Sport and Health Sciences, Cardiff Metropolitan University, Cardiff CF23 6XD, UK; jpedley@cardiffmet.ac.uk (J.S.P.); joliver@cardiffmet.ac.uk (J.L.O.); rlloyd@cardiffmet.ac.uk (R.S.L.); 2Sport Performance Research Institute New Zealand, AUT University, Auckland 0632, New Zealand; 3Centre for Sport Science and Human Performance, Waikato University of Technology, Hamilton 3240, New Zealand

**Keywords:** drop jump, kinetics, force, gymnastics, girls

## Abstract

This cross-sectional study investigated how stretch-shortening cycle (SSC) function and kinetic variables differed between young female gymnasts of varying competitive levels. Drop jump (DJ) force–time profiles were examined in 118 female gymnasts, sub-divided by competitive level (*n* = 21 recreational, *n* = 41 regional and *n* = 50 elite). DJ force–time data were analyzed to calculate performance and kinetic variables. Participants’ SSC function was categorized as poor, moderate, or good, depending on the presence of an impact peak and spring-like behavior. A high proportion of gymnasts across each group were categorized as having “good” or “moderate” SSC function (i.e., >94.8%), with a trend of increasingly better SSC function observed with competitive level. Significant differences in reactive strength index, contact time, time of landing peak force, relative propulsive peak force, impulse, and ratio of braking: propulsive impulse were found between the elite and recreational group (*p* < 0.05). While SSC function was generally good to moderate, elite gymnasts had a more desirable kinetic jump-landing strategy than recreational level gymnasts. Drop jump kinetic variables appear to distinguish between elite and recreational gymnasts but not between regional standard gymnasts. Practitioners should consider the kinetic profile of gymnasts when benchmarking and setting training objectives.

## 1. Introduction

For female artistic gymnasts, successful performance in three of the four disciplines (i.e., vault, beam and floor) are in part governed by a gymnasts’ ability to perform technically demanding skills while airborne [1]. Therefore, there is a need to maximize the duration of the aerial phase to provide more time to execute complex, higher scoring skills. The ability to jump and rebound off the ground, or apparatus such as the beam or vault, requires effective utilization of the stretch-shortening cycle (SSC). The SSC involves a rapid sequence of eccentric, isometric and concentric muscle actions, with the initial eccentric braking phase augmenting performance in the subsequent concentric propulsive phase [2]. Within the gymnastics literature, SSC function has typically been assessed using various jumping and rebounding assessments in order to physically profile young gymnasts and identify key determinants of gymnastics performance [3,4,5]. Existing research has shown that jump height, maximal vertical force, and maximal and mean power are significantly greater in gymnasts aged 12–16 years, in comparison to gymnasts aged 9–12 years [5]. Additionally, increased age, faster run-up speed and shorter ground contact times were identified as a key determinant of tumbling ability in a cohort of gymnasts aged 8–14 years [6].

Current literature shows that gymnasts typically outperform their non-gymnast peers on a range of jump tests, including squat and countermovement jumps [7] and drop jumps [7,8]. These findings may reflect how participation in the sport of gymnastics can serve as a vehicle to improve indices of strength and power performance [9], or alternatively coaches intuitively select gymnasts with good jumping abilities. Recently, research has shown that rate-of-force development (RFD) during an isometric mid-thigh pull is significantly greater in elite level young female gymnasts compared to recreational level gymnasts [10]. Within the study, elite level gymnasts were participating in approximately four times the amount of gymnastics training compared to the recreational gymnasts (18.9 ± 4.0 h·wk^−1^ vs. 4.4 ± 1.8 h·wk^−1^). Intuitively, these results would indicate that higher training loads and exposure to more advanced skills, which necessitate more forceful and higher velocity muscle actions, provide a training stimulus that can augment explosive force production. However, how competitive level influences kinetic variables in jump protocols that stress the SSC remains unclear.

One of the primary limitations of the existing pediatric gymnastics literature has been an over-reliance on SSC testing protocols that use field-based equipment such as contact mats [3,11] and/or solely provide performance-based variables, for example countermovement jump height [12,13]. Such protocols provide surrogate measures of SSC function; however, kinetic analysis provides greater insight into the strategies that dictate the eventual performance (i.e., height jumped). For example, reactive strength index (RSI) is a commonly reported variable when using the drop jump for performance testing [14]. However, given RSI is a ratio determined from jump height and ground contact time, it can be increased through a potentially undesirable strategy (e.g., increased jump height but at the expense of a prolonged ground contact time). Further, data indicate that performance variables such as RSI may not possess the sensitivity to detect alterations in kinetic strategies used during a drop jump [15]. Recent drop jump kinetic data has shown that late prepubertal and pubertal female gymnasts jumped significantly higher than early prepubertal gymnasts by producing significantly greater amounts of peak force, work, power and relative stiffness [16]. However, the manner in which drop jump kinetics differ between young female gymnasts of different competitive levels remains unknown.

Recently, a novel method of categorizing SSC function in young athletes has been proposed, which considers whether there is an impact peak in vertical force during the early stage of ground contact and if the individual displays spring-like behavior during the task [17]. Using this categorization approach, recent data has shown that in a cohort of 1013 young females, the majority of girls (~80% prepubertal, ~66% postpubertal) displayed poor SSC function, characterized by landing peaks and a lack of spring-like behavior, irrespective of maturity status [18]. Interestingly, Pedley et al. [18] also reported that while performance variables (e.g., RSI) remained relatively consistent across different stages of maturity, the underlying kinetic strategies changed with advancing maturity. In addition to considering maturity status, having access to benchmark data for drop jump kinetics at different competitive levels of gymnastics will enable practitioners to better interpret SSC function of young gymnasts and assist with programming decisions.

In light of the current evidence, the aims of the current study were to firstly determine how SSC function varied across different competitive levels of gymnastics and secondly, to provide benchmark data for key SSC variables for recreational, regional and elite/national gymnasts. For the first aim, two research hypotheses were tested: (1) the relative proportion of gymnasts categorized as displaying “good” SSC function would improve with competitive level; (2) these differences would be underpinned by more efficient kinetic strategies (e.g., a higher spring-like correlation, greater peak force and impulse, and a shorter ground contact time, etc.).

## 2. Materials and Methods

### 2.1. Study Design

This study used a cross-sectional design to examine ground reaction force variables as indicators of SSC function in artistic female gymnasts who were grouped according to competitive level. All subjects attended one testing session in which anthropometric and DJ performance data were collected.

### 2.2. Participants

One hundred and eighteen female artistic gymnasts aged 5–14 years volunteered to participate in the study. All participants were from gymnastics clubs in South Wales and had >1 years of gymnastics experience and were participating in gymnastics training 2–6 times per week, totaling 2–24 training hours per week. Participants were grouped according to their competitive level of gymnastics: Elite/National (*n* = 50), Regional (*n* = 47), and Recreational (*n* = 21). Competitive levels were defined by the classifications presented in Table 1. Participants biological maturity was also estimated using percentage of predicted adult height attained (%PAH) [19] using the following categories: <75% PAH, early prepubertal; 76–85% PAH, late prepubertal; and 86–95% PAH, pubertal [16]. Descriptive data for the participants are shown in Table 2. Participants reported no injuries at the time of testing and were instructed to refrain from strenuous activity 24 h before testing. Ethical approval for the study was granted by the ethics board at Cardiff Metropolitan University (Ethics code: 17/1/02R). Participants and parents were informed of the benefits and risks of the investigation before signing institutionally approved informed assent and consent documents.

### 2.3. Procedures

#### 2.3.1. Familiarization

Prior to testing, all participants performed a standardized 10-min dynamic warm-up led by the principal researcher, including relevant activation and mobilization exercises and 3 sets of squat jumps, countermovement jumps, and pogo hops. Familiarization of the DJ protocol took place before the testing session commenced. This involved the principal researcher providing a demonstration of the DJ and standardized child-friendly coaching cues. Participants then practiced the protocol until the researcher was satisfied with the gymnasts’ technical competency.

#### 2.3.2. Anthropometrics

Anthropometric data were collected, including standing and sitting height using a stadiometer to the nearest 0.1 cm (SECA, 321, Vogel & Halke, Hamburg, Germany) and body mass using scales to the nearest 0.1 kg (SECA, 321, Vogel & Halke, Hamburg, Germany). Standing height (cm), body mass (kg), chronological age and parental height were used to determine participants’ biological maturity status, using %PAH [19].

#### 2.3.3. Drop Jump

Data were collected in a laboratory using two force plates sampling at a frequency of 1000 Hz (PASCO, 2 Axis force platforms, Roseville, CA, USA). The DJ protocol required the participants to step out and off a 30 cm platform (positioned 10 cm from the contact area), land on two force plates, and rebound as high as possible with a fast ground contact time [16]. Participants were cued to “step out off of the box and rebound as high and as fast as possible” [16]. Gymnasts were instructed to keep their hands on their hips throughout and keep their legs extended during the flight phase of the jump. Trials where the gymnasts noticeably stepped down or jumped up from the platform were discounted and repeated [20]. The gymnasts were afforded a minimum of 60 s passive rest between trials, to enable sufficient recovery [16]. Three trials were completed, with the best of the 3 trials used for further analyses. The best trial was determined by the highest spring-like behavior correlation (i.e., a perfect inverse relationship is indicated by *r* = −1.0), which represents spring-mass model behavior [17]. Note, spring-like behavior is determined by a correlation of *r* > −0.80 between vertical center of mass displacement (∆COM) and absolute vertical force, throughout the first ground contact [17].

All DJ data were analyzed using a bespoke Excel Spreadsheet. Ground reaction force data was smoothed using a fourth order recursive low-pass Butterworth filter with a cut-off of 30 Hz [21]. Touchdown was determined by the first time sample recording of ground reaction force > 15 N and take-off was identified as the next time sample of force < 15 N [17]. Once the data had been filtered, the program analyzed the data to determine the following performance variables: jump height, ground contact time (CT) and reactive strength index (RSI). RSI was determined by dividing jump height (m) by the first ground contact period(s). The mechanical variables calculated included: ∆COM, spring-like correlation, relative peak landing force, time of landing peak force, relative peak take-off force, net impulse, braking impulse, propulsive impulse, braking duration, propulsive duration, and the ratio of braking: propulsive impulse. All relative measures were calculated using body mass to increase the utility of the bench-mark data. The within-session reliability of the DJ variables were found to have mean coefficients of variation between 3–15% [22]. Further information on the variables calculated can be found in the Appendix A. The gymnasts’ SSC function were categorized using methods by Pedley et al. [17], which classifies individuals based on whether an impact peak (the highest visible force peak during the first 20% of the landing phase of ground contact) and spring-like correlation (spring-like behavior ≥ −0.80) is observed or not [23]. The following classifications were used: “good”; no impact peak and spring-like, “moderate”; impact peak but spring-like, or “poor”; impact peak and not spring-like [17].

### 2.4. Statistical Analyses

Descriptive statistics (mean values ± SD) were calculated for all anthropometric and DJ data for each competitive group. Differences in variables between competitive groups were assessed using analysis of covariance (ANCOVA) to control for biological maturity. Participants %PAH was used as a covariate and Bonferroni post hoc test to identify pairwise comparisons. Homogeneity of variance was assessed using Levene’s statistic and, where violated, Welch’s adjustment was used to correct the F-ratio. Effect sizes (Cohen’s *d*) were also calculated to establish the magnitude of between-group differences [24] using the following classifications: <0.2, trivial; 0.2–0.59, small; 0.6–1.19, moderate; 1.2–1.99, large; 2.0–4.0, very large; and >4.0, nearly perfect [25]. The 10th, 50th and 90th percentiles were calculated for all DJ variables for each competitive group. All significance values were accepted at *p* < 0.05, and all statistical procedures were conducted using SPSS v.24 for MacIntosh (IBM, New York, NY, USA).

## 3. Results

### 3.1. Anthropometrics

Mean ± SD data for anthropometrics are shown in Table 1. Data showed the recreational group were significantly taller than both the regional and elite/national group (*p* < 0.05). The recreational group also had a significantly greater body mass than the elite/national group (*p* < 0.05). However, no significant differences were found between groups for maturity status (i.e., %PAH).

### 3.2. Drop Jump

The proportion of gymnasts categorized as having “good”, “moderate” or “poor” SSC function within each competitive level group is displayed in Figure 1. Overall, a high proportion of gymnasts across each group were categorized as having “good” or “moderate” SSC function (i.e., >94.8%). A trend of increasingly better SSC function with competitive level was also observed (e.g., the proportion of gymnasts with “good” SSC = recreational, 71.4%; regional = 87.2%; and elite/national = 88.0%).

Results for DJ performance variables are shown in Figure 2. Data revealed the elite/national group had a significantly greater RSI (*p* = 0.03, *d* = 0.64) and a significantly shorter CT (*p* = 0.03, *d* = −0.78) than the recreational (Figure 2). No significant differences were also observed between the recreational and regional group for RSI or CT, but effect sizes were small to moderate (*d* = −0.54–0.67). For jump height data, small non-significant effect sizes were found between both higher-level competitive groups and the recreational group (*d* = 0.24–0.38). However, no significant differences were found between the regional and elite/national group for any DJ performance variables, and effect sizes were trivial to small (*d* = −0.06–0.26).

Group results for DJ kinetic variables are shown in Table 3. The elite/national group had a significantly greater time of landing peak force and relative peak propulsive force than the recreational group (*p* = 0.02, *d* = 0.80 and *p* = 0.01, *d* = 0.87, respectively). No significant differences were found among groups for: peak force, relative peak force, ∆COM displacement, spring-like correlation, or relative peak braking force (*d* = 0.08–0.59). However, small effect sizes were evident for relative peak force between both higher-level competitive groups and the recreational group (*d* = 0.56–0.58). Analysis from the impulse variables revealed the recreational group had a significantly greater braking and propulsive impulse, and longer propulsive impulse duration than the elite/national group (*p* = 0.00–0.02, *d* = 0.46–0.89). However, no significant differences were found between any groups for net impulse or braking phase duration (*d* = −0.08–0.52). Data also showed the elite/national group had a significantly greater ratio of braking: propulsive impulse than the recreational group (*p* = 0.01, *d* = 0.65). No significant differences were found between the regional and elite/national group for any DJ kinetic variables (*d* = −0.13–0.30).

Benchmark data using the 10th, 50th and 90th percentile for each competitive group is shown in Table 4 for RSI, jump height, CT, spring-like correlation, and the ratio of braking: propulsive impulse.

## 4. Discussion

In partial agreement with hypothesis one, the proportion of gymnasts displaying “good” SSC function was greater in regional standard gymnasts compared to recreational, though there was no difference in proportion displaying “good” SSC function between regional and elite competitive gymnasts. In agreement with hypothesis two, there were trends for all kinetic variables to improve with competitive level with many variables showing statistically significant differences between recreational and elite competitors.

The SSC enhances both the magnitude of, and the time required to produce, a given amount of impulse [26]. This reduced time to produce impulse facilitates greater sprinting speed [27], which has been demonstrated to be necessary for execution of more demanding vault skills in female gymnasts [28]. Enhanced SSC performance will also facilitate greater aerial times during floor skills and apparatus dismounts to allow execution of more complex technical skills [29,30]. The current study observed that SSC function increases with competitive level in gymnasts consolidating these previous findings and emphasizing the importance of SSC development for youth gymnasts. Furthermore, examining SSC function via the DJ protocol could be utilized as a potential talent development tool. For example, utilizing percentile thresholds (i.e., 10th, 50th and 90th) for key kinetic variables could help to profile young female gymnasts’ SSC function, monitor training progress and to distinguish between competitive levels. For example, an elite level gymnast who scores in the 90th percentile for relative peak propulsive force but is in the 10th percentile for propulsive duration should arguably prioritize training to develop rate-of-force-development. Consequently, practitioners could then use select training modes such as plyometrics and weightlifting derivatives to target these specific physiological adaptations.

Previous researchers have reported that exposure to gymnastics-specific training could serve as an effective stimulus for SSC development [9]. While there were consistent trends for regional gymnasts to outperform their recreational counterparts and for elite gymnasts to outperform regional gymnasts, there were only significant differences between elite and recreational groups. The mean training time for the elite gymnasts in the current study was 243% greater than the recreational group. This observation suggests that while there might be a dose–response relationship between gymnastics exposure and the extent of SSC development, large training volumes are required to provide a sufficient stimulus to significantly improve SSC function, and thus supplementary strength and conditioning training might be appropriate, particularly in lower competitive levels.

The majority (~71%) of recreational gymnasts in the current study displayed “good” SSC function, while this number was even greater in regional and elite/national standard gymnasts (87–88%). In a sample of >1000 female basketball, volleyball and soccer players of mixed maturity status, the proportion of athletes with “good” SSC function was observed to be <10% [18]. Such a discrepancy is likely to be partly due to a selection bias for the sport of gymnastics whereby a baseline level of SSC function is required to be successful even at a recreational level. Notably, the specificity of the gymnastics-training stimulus itself may also enhance SSC function (e.g., fast SSC actions occurring in rebounding and tumbling, etc.). Therefore, gymnastics talent development screening batteries should include fast SSC assessments such as the drop jump. The current study is the first to provide benchmark data for key drop jump kinetic variables that distinguish between competitive standards.

RSI is a common metric to report in drop jump assessments, and the current data set illustrate that it is a distinguishing characteristic between recreational and elite gymnasts. However, it is important to understand the strategy utilized to achieve a given RSI since it is a ratio variable [31]. There was no difference in jump height between any competitive level in this study; the greater RSI observed in the elite level gymnasts was achieved because of shorter ground contact times and thus any screening assessment should also report ground contact time alongside RSI, since the same RSI could be achieved by a jump height dominant strategy at the expense of ground contact time.

Previous researchers have emphasized the importance of practitioners’ understanding the kinetic strategies that underpin the eventual performance outcome (i.e., height jumped) [16]. Drop jump kinetic variables were significantly different between elite and recreational gymnasts. With advancing competitive level, the timing of relative peak braking force becomes later during the ground contact period, moving closer to the mid-point as would be observed in perfect spring-like behavior [17,32]. When peak braking force occurs in the first 20% of ground contact time, worse performance outcomes have been observed which has been speculated to be due to ineffective dampening mechanisms and impaired utilization of stored elastic energy [17,33]. Higher level gymnasts in this study appear to have better dampening mechanisms which could facilitate better storage of elastic energy into connective tissues. This is supported by the observation that elite gymnasts had significantly shorter propulsive durations and greater relative peak propulsive force in comparison to recreational gymnasts. Force would be higher at the initiation of the propulsive phase meaning the necessary impulse for take-off could be achieved in a shorter duration. This kinetic strategy would facilitate the significantly shorter ground contact times observed in elite standard competitors in comparison to recreational gymnasts.

There were significant differences in both braking and propulsive impulse between recreational and elite level gymnasts. However, there was no significant difference in net impulse between any of the groups, which indicates these differences were largely down to the significant differences in body mass between the two groups. It also supports the notion that in rebounding SSC tasks involving a collision between the athlete and landing surface, absolute impulse is less important than the distribution of that impulse throughout the ground contact period [17,18].

Certain limitations should be noted for this study. SSC function was only evaluated during a drop jump which involves a fast-SSC (i.e., ground contact times less than 250 ms) however, gymnasts often also utilize slow-SSC actions during jumping and acrobatic skills. There were differences in sample sizes of the subgroups, with the recreational group having less participants than the other two groups overall. Despite these limitations, the current study makes a novel and significant contribution to the pediatric and gymnastics literature, indicating that SSC function during a drop jump improves with competitive level and that these gymnasts had a more desirable jump-landing strategy.

## 5. Conclusions and Practical Applications

The present study is the first to profile drop jumping ability in different standard female gymnasts. Elite standard gymnasts displayed a higher proportion of “good” SSC function, reflected by significantly greater RSI scores. The heightened SSC function was achieved through a ground contact time focused jump-landing strategy underpinned by peak braking force occurring later during ground contact and a shorter propulsive duration. Drop jump kinetic variables appear to distinguish between elite and recreational gymnasts and could be used to help practitioners’ set training objectives.

## Figures and Tables

**Figure 1 sports-10-00107-f001:**
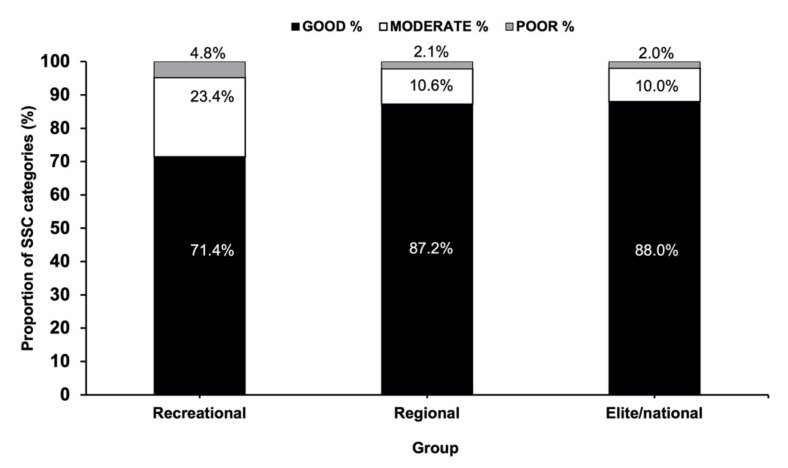
Proportions of gymnasts categorized as having “good”, “moderate” or “poor” SSC function across different competitive level groups.

**Figure 2 sports-10-00107-f002:**
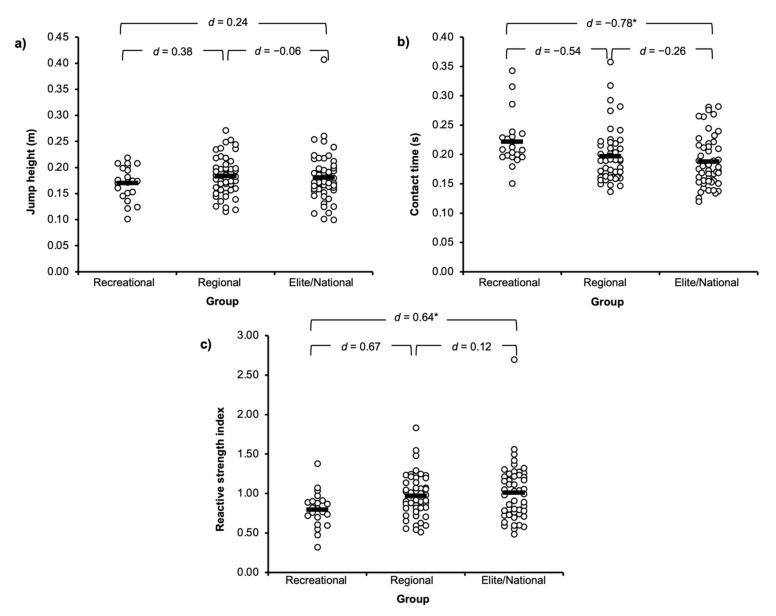
Competitive level group analysis (mean ± sd) for (**a**) jump height; (**b**) ground contact time; and (**c**) reactive strength index. * = significant difference; *d* = Cohen’s d effect size.

**Table 1 sports-10-00107-t001:** Group definitions for competitive levels of gymnastics.

Title 1	Title 2
Recreational	Gymnasts who have not participated in grades and have not been identified to compete at this level or above
Regional	Gymnasts who have competed in regional grades or have been identified to potentially compete at this level
Elite/national	Gymnasts who have competed in national or compulsory elite grades or those who have been identified to potentially compete at this level

**Table 2 sports-10-00107-t002:** Descriptive statistics for age, maturity status and anthropometric variables (mean ± SD).

Group	*N*	Age (Years)	Standing Height (cm)	Body Mass (Kg)	Predicted % Adult Height	Training Hours per Week
Recreational	21	9.6 ± 2.6	144.74 ± 7.56 ^ab^	33.5 ± 11.6 ^a^	82.1 ± 8.2	4.4 ± 1.8
Regional	47	9.8 ± 1.8	130.74 ± 11.50	31.8 ± 8.7	82.8 ± 7.0	9.8 ± 3.1
Elite/national	50	9.7 ± 2.1	133.57 ± 12.75	30.1 ± 7.7	82.2 ± 6.9	15.1 ± 4.3 ^bc^

^a^ = Significantly greater than the elite/national group; ^b^ = Significantly greater than the regional group; ^c^ = Significantly greater than the recreational group; Significant at the level of *p* < 0.05.

**Table 3 sports-10-00107-t003:** Level of competition group analysis for drop jump variables.

	Means ± SD			Effect Size Cohen’s *d* and Confidence Intervals		
	Recreational	Regional	Elite/National	Recreational vs. Regional	Regional vs. Elite/National	Recreational vs. Elite/National
Peak force (N)	1812.39 ± 590.55	1952.20 ± 536.40	1874.25 ± 516.74	0.25 (−0.27 to 0.77)	−0.15 (−0.55 to 0.15)	0.11 (−0.40 to 0.62)
Relative peak force (BW)	5.61 ± 1.13	6.36 ± 1.37	6.52 ± 1.76	0.58 (0.05 to 1.09)	0.10 (−0.30 to 0.47)	0.56 (0.04 to 1.08)
∆COM displacement (m)	−0.11 ± 0.03	−0.11 ± 0.03	−0.11 ± 0.03	0.10 (−0.42 to 0.61)	0.10 (−0.30 to 0.50)	0.20 (−0.31 to 0.71)
Spring-like Correlation (*r*)	−0.90 ± 0.05	−0.92 ± 0.05	−0.93 ± 0.05	−0.25 (−0.77 to 0.27)	−0.24 (−0.64 to 0.16)	−0.48 (−0.99 to 0.04)
Relative peak braking force (BW)	5.60 ± 1.14	6.36 ± 1.37	6.49 ± 1.17	0.59 (0.06 to 1.10)	0.08 (−0.32 to 0.48)	0.55 (0.03 to 1.06)
Time of landing peak force (%)	23. 19 ± 6.17	26.22 ± 6.19	28.68 ± 7.17 ^a^	0.49 −0.04 to 1.01)	0.37 (−0.04 to 0.77)	0.80 (0.26 to 1.32)
Relative peak propulsive force (BW)	4.22 ± 0.79	4.89 ± 1.05	5.37 ± 1.48 ^a^	0.69 (0.15 to 1.21)	0.37 (−0.03 to 0.77)	0.87 (0.33 to 1.39)
Net impulse (N·s)	130.49 ± 40.46	127.38 ± 38.10	118.65 ± 31.33	−0.08 (−0.59 to 0.44)	−0.25 (−0.65 to 0.15)	−0.35 (−0.58 to 0.17)
Braking impulse (N·s)	95.34 ± 32.93 ^b^	90.35 ± 30.45	83.36 ± 22.94	−0.16 (−0.67 to 0.36)	−0.26 (−0.66 to 0.14)	−0.46 (−0.97 to 0.06)
Propulsive impulse (N·s)	109.75 ± 35.93 ^b^	101.60 ± 33.18	92.96 ± 24.31	−0.24 (−0.67 to 0.36)	−0.30 (−0.53 to 0.15)	−0.60 (−1.11 to −0.07)
Braking duration (s)	0.09 ± 0.02	0.08 ± 0.02	0.08 ± 0.02	−0.35 (−1.15 to −0.10)	−0.13 (−0.53 to 0.27)	−0.52 (−1.03 to 0.00)
Propulsive duration (s)	0.13 ± 0.03 ^b^	0.12 ± 0.03	0.11 ± 0.03	−0.63 (−1.15 to −0.10)	−0.25 (−0.65 to 0.15)	−0.89 (−1.41 to −0.35)
Ratio of braking: propulsive impulse	0.86 ± 0.06	0.89 ± 0.04	0.89 ± 0.04 ^a^	0.49 (−0.04 to 1.01)	0.17 (−0.55 to 0.25)	0.65 (0.12 to 1.16)

Significant at the level of *p* < 0.05. ^a^ = significantly greater than recreational group; ^b^ = significantly greater than the elite/national group. Small effect size: 0.20 to 0.59, moderate effect size: 0.60 to 1.19, and large effect size: 1.20 to 1.99.

**Table 4 sports-10-00107-t004:** Benchmark data percentiles for each competitive group.

	Recreational			Regional			Elite/National		
Variable	10th	50th	90th	10th	50th	90th	10th	50th	90th
RSI	0.55	0.81	1.04	0.62	0.98	1.25	0.60	1.01	1.33
Jump height (m)	0.12	0.17	0.21	0.14	0.18	0.24	0.13	0.17	0.23
Contact time (s)	0.29	0.21	0.19	0.26	0.19	0.16	0.27	0.18	0.14
Peak force (N)	1147	1749	2296	1325	1872	2617	1214	1797	2556
Relative peak force (BW)	4.23	5.78	6.99	4.86	6.36	7.88	4.54	6.28	8.52
∆COM displacement (m)	−0.15	−0.10	−0.09	−0.15	−0.10	−0.08	−0.14	−0.10	−0.07
Spring-like correlation (*r*)	−0.86	−0.87	−0.97	−0.87	−0.90	−0.97	−0.86	−0.94	−0.98
Relative peak braking force (BW)	4.23	5.78	7.00	4.86	6.36	7.88	4.46	6.28	8.52
Time of landing peak force (%)	15.74	22.96	28.33	18.21	25.86	32.66	19.59	28.77	38.10
Relative peak propulsive force (BW)	3.42	4.34	4.70	3.69	4.78	6.20	3.81	5.20	6.66
Net impulse (N·s)	87.02	123.45	173.36	84.16	120.38	171.12	77.03	117.79	154.94
Braking impulse (N·s)	65.60	95.30	134.44	58.04	85.32	122.37	52.82	84.95	103.77
Propulsive impulse (N·s)	74.57	100.67	153.59	65.60	95.30	134.44	62.39	94.83	117.95
Braking duration (s)	0.11	0.08	0.07	0.10	0.08	0.06	0.11	0.07	0.05
Propulsive duration (s)	0.16	0.13	0.11	0.15	0.11	0.90	0.15	0.10	0.08
Ratio of braking: propulsive impulse	0.82	0.87	0.92	0.83	0.89	0.94	0.84	0.89	0.95

## Data Availability

Research data are not shared due to privacy and ethical restrictions.

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
