# Peer review of "The Influence of Competitive Level on Stretch-Shortening Cycle Function in Young Female Gymnasts"

_sports, 2022, doi:10.3390/sports10070107_

Round 1

Reviewer 1 Report

The study investigates how stretch-shortening cycle (SSC) function and kinetic variables differ between young female gymnasts of varying competitive levels. The authors concluded that drop jump kinetic variables appear to distinguish between elite and recreational gymnasts but not between regional standard gymnasts.

Findings of this study are in my opinion of relevance to sport science and would fit the scope of the journal. There are, however, minor concerns which should be addressed.

Please specify “efficient kinetic strategies”.

L91-94: For the first aim, two research hypotheses were tested: 1) the relative proportion of gymnasts categorized as displaying “good” SSC function would improve with competitive level; 2) these differences would be underpinned by more efficient kinetic strategies.

Please, include p values.

L195-197: Data revealed the elite/national group had a significantly greater RSI (p < 0.05, d = 0.64) and a significantly shorter CT (p < 0.05, d = -0.78) than the recreational (Figure 2).

L204-206: The elite/national group had a significantly greater time of landing peak force and relative peak propulsive force than the recreational group (p < 0.05, d = 0.80 and d = 0.87, respectively).

L210-213: Analysis from the impulse variables revealed the recreational group had a significantly greater braking and propulsive impulse, and longer propulsive impulse duration than the elite/national group (p < 0.05, d = 212 0.46-0.89).

L214-216: Data also showed the elite/national group had a significantly greater ratio of braking: propulsive impulse than the recreational group (p < 0.05, d = 0.65).

I suggest removing the aim of the study from the “Discussion”

L239-240: The aim of the present study was to identify SSC function at different competitive levels of gymnastics.

Relevant limitations of this research should be discussed.

Please reformulate this sentence (the usage of drop jump kinetic variables for talent identification has not been investigated in the present study)

L319-320: Drop jump kinetic variables appear to distinguish between elite and recreational gymnasts and could be used for talent identification and setting training objectives.

This part should be included in the “Discussion”. Practical applications of findings with respect to young female gymnasts that are not currently addressed in the literature should also be included.

 L321-326: Benchmark data are provided for key kinetic variables to aid practitioners in these endeavors. For example, an elite level gymnast who scores in the 90th percentile for relative peak propulsive force but is in the 10th percentile for propulsive duration should arguably prioritize training to develop rate-of-force-development. Consequently, practitioners could then use select training modes such as plyometrics and weightlifting derivatives to target these specific physiological adaptations.

Author Response

The study investigates how stretch-shortening cycle (SSC) function and kinetic variables differ between young female gymnasts of varying competitive levels. The authors concluded that drop jump kinetic variables appear to distinguish between elite and recreational gymnasts but not between regional standard gymnasts.

Findings of this study are in my opinion of relevance to sport science and would fit the scope of the journal. There are, however, minor concerns which should be addressed.

Response: Thank you for taking the time to review our paper. We feel that the comments and feedback provided have overall improved the quality of our work.

Please specify “efficient kinetic strategies”.

L91-94: For the first aim, two research hypotheses were tested: 1) the relative proportion of gymnasts categorized as displaying “good” SSC function would improve with competitive level; 2) these differences would be underpinned by more efficient kinetic strategies.

Response: Many thanks for your comment, we have added the following text.  “2) these differences would be underpinned by more efficient kinetic strategies (e.g. a higher spring-like correlation, greater peak force and impulse, and a shorter ground contact time etc).” (lines 94-95)

Please, include p values.

L195-197: Data revealed the elite/national group had a significantly greater RSI (< 0.05, = 0.64) and a significantly shorter CT (< 0.05, = -0.78) than the recreational (Figure 2).

L204-206: The elite/national group had a significantly greater time of landing peak force and relative peak propulsive force than the recreational group (< 0.05, = 0.80 and = 0.87, respectively).

L210-213: Analysis from the impulse variables revealed the recreational group had a significantly greater braking and propulsive impulse, and longer propulsive impulse duration than the elite/national group (< 0.05, = 212 0.46-0.89).

L214-216: Data also showed the elite/national group had a significantly greater ratio of braking: propulsive impulse than the recreational group (< 0.05, = 0.65).

Response: Thank you, we have added the p values (lines 200-221).

I suggest removing the aim of the study from the “Discussion”

L239-240: The aim of the present study was to identify SSC function at different competitive levels of gymnastics.

Response: Many thanks for your comment, we have removed this sentence.

Relevant limitations of this research should be discussed.

Response: The authors welcome this comment, and the following text has been added. ”Certain limitations should be noted for this study. SSC function was only evaluated during a drop jump which involves a fast-SSC (i.e. ground contact times less than 250 ms) however, gymnasts often also utilize slow-SSC actions during jumping and acrobatic skills. There were differences in sample sizes of the subgroups, with the recreational group having less participants than the other two groups overall. Despite these limitations, the current study makes a novel and significant contribution to the pediatric and gymnastics literature, indicating that SSC function during a drop jump improves with competitive level and that these gymnasts had a more desirable jump-landing strategy.” (lines 320-328).

Please reformulate this sentence (the usage of drop jump kinetic variables for talent identification has not been investigated in the present study)

L319-320: Drop jump kinetic variables appear to distinguish between elite and recreational gymnasts and could be used for talent identification and setting training objectives.

Response: We have amended the sentence to read “Drop jump kinetic variables appear to distinguish between elite and recreational gymnasts and could be used to help practitioners’ set training objectives.” (lines 335-336)

This part should be included in the “Discussion”. Practical applications of findings with respect to young female gymnasts that are not currently addressed in the literature should also be included.

L321-326: Benchmark data are provided for key kinetic variables to aid practitioners in these endeavors. For example, an elite level gymnast who scores in the 90th percentile for relative peak propulsive force but is in the 10th percentile for propulsive duration should arguably prioritize training to develop rate-of-force-development. Consequently, practitioners could then use select training modes such as plyometrics and weightlifting derivatives to target these specific physiological adaptations.

Response: The authors thank you for the comment and have moved the following text to the discussion (lines 261-266) “For example, an elite level gymnast who scores in the 90th percentile for relative peak propulsive force but is in the 10th percentile for propulsive duration should arguably prioritize training to develop rate-of-force-development. Consequently, practitioners could then use select training modes such as plyometrics and weightlifting derivatives to target these specific physiological adaptations.”  

Reviewer 2 Report

The authors investigated the differences in stretch-shortening cycle function between three competitive levels of female gymnasts by performing a drop jump protocol. The authors were also able to present normative data for the three competitive levels across 12 mechanical variables that were recorded during the drop jump performance, thereby providing practitioners with data to potentially use for talent identification and establishing training objectives.

The manuscript is well written and provides an approach to assessing stretch-shortening cycle function that is fairly new, but adds to some recent data sets. The practical applications from the data provided by the authors will undoubtedly be useful to practitioners involved with young female gymnasts. My comments are minimal and largely revolve around clarification of elements of the Methods section. The authors should be congratulated on an interesting study that is well presented.

Specific comments

Materials and Methods, page 3, line 109: Consider changing ‘bands’ to ‘categories’.

Materials and Methods, page 3, lines 147-149: Clarify here for the reader that the appearance of spring-like behavior is determined by a correlation of > -0.80 between the vertical displacement of the COM and the vertical GRF during the contact phase of the DJ. This is not clear in your current document.

Materials and Methods, page 3, line2 153-154: Clarify for the reader how the events of touchdown and take-off were determined from the GRF data. Also, provide an equation for the calculation of RSI here. I appreciate that there will be greater detail presented in the Supplementary Data section, but I think that it would be useful just to present a brief description of these variables here.

Materials and Methods, page 3, line 154: Consider changing ‘kinetic variables’ to ‘mechanical variables’ as displacement is not a kinetic variable (also required in line 204 on page 5).

Results, page 5, line 209: Change to ‘Peak force’

Discussion, page 8, line 259: Change to “Previous researchers have reported...”

Discussion, page 9, line 289: Change to “Previous researchers have emphasized...”

Table 3: The meaning of ‘a’ in the statistical analyses is not presented in the note below the table.

Table 4: Consider writing-out contact time rather than CT as the space is available for this.

Author Response

The authors investigated the differences in stretch-shortening cycle function between three competitive levels of female gymnasts by performing a drop jump protocol. The authors were also able to present normative data for the three competitive levels across 12 mechanical variables that were recorded during the drop jump performance, thereby providing practitioners with data to potentially use for talent identification and establishing training objectives.

The manuscript is well written and provides an approach to assessing stretch-shortening cycle function that is fairly new, but adds to some recent data sets. The practical applications from the data provided by the authors will undoubtedly be useful to practitioners involved with young female gymnasts. My comments are minimal and largely revolve around clarification of elements of the Methods section. The authors should be congratulated on an interesting study that is well presented.

Response: The authors would like to thank the reviewer for their time and expertise. We feel that the paper has been improved following their review.

Specific comments

Materials and Methods, page 3, line 109: Consider changing ‘bands’ to ‘categories’.

Response: Thank you for your comment, we have amended this text (line 110).

Materials and Methods, page 3, lines 147-149: Clarify here for the reader that the appearance of spring-like behaviour is determined by a correlation of > -0.80 between the vertical displacement of the COM and the vertical GRF during the contact phase of the DJ. This is not clear in your current document.

Response: We welcome the reviewer’s comment and have added the following text: “Note, spring-like behavior is determined by a correlation of r > -0.80 between vertical centre of mass displacement (∆COM) and absolute vertical force throughout the first ground contact [17].” (lines150-152)

Materials and Methods, page 3, line2 153-154: Clarify for the reader how the events of touchdown and take-off were determined from the GRF data. Also, provide an equation for the calculation of RSI here. I appreciate that there will be greater detail presented in the Supplementary Data section, but I think that it would be useful just to present a brief description of these variables here.

Response: Thank you for your comments, we have added text and the amended paragraph is as follows “Touchdown was determined by the first time sample recording of ground reaction force >15N and take-off was identified as the next time sample of force <15N [17]. Once the data had been filtered, the program analyzed the data to determine the following performance variables: jump height, ground contact time (CT) and reactive strength index (RSI). RSI was determined by dividing jump height (m) by the first ground contact period (s).” Lines 155-160.

Materials and Methods, page 3, line 154: Consider changing ‘kinetic variables’ to ‘mechanical variables’ as displacement is not a kinetic variable (also required in line 204 on page 5).

Response: This is a valid point, thank you (amended line 160).

Results, page 5, line 209: Change to ‘Peak force’

Response: Thank you for the thorough review, we have amended the text (line 214).

Discussion, page 8, line 259: Change to “Previous researchers have reported...”

Response: We have amended the text as suggested, thank you (line 267).

Discussion, page 9, line 289: Change to “Previous researchers have emphasized...”

Response: Amended, thank you (line 297).

Table 3: The meaning of ‘a’ in the statistical analyses is not presented in the note below the table.

Response: Amended, thank you (page 7).

Table 4: Consider writing-out contact time rather than CT as the space is available for this.

Response: Amended (page 8).